# Raman Scattering in a Double-Doped Single Crystal LiTaO$_3$:Cr(0.2):Nd(0.45 wt%)

**Nikolay Sidorov** [1,*], **Mikhail Palatnikov** [1] and **Alexander Pyatyshev** [2]

1    Tananaev Institute of Chemistry–Subdivision of the Federal Research Centre «Kola Science Centre of the Russian Academy of Sciences», 184209 Apatity, Russia

2    P.N. Lebedev Physical Institute of the Russian Academy of Sciences, 119991 Moscow, Russia

*    Correspondence: n.sidorov@ksc.ru

**Abstract:** The Raman spectra of a lithium tantalate crystal doubly doped with chromium and neodymium LiTaO$_3$:Cr(0.2):Nd(0.45 wt%) have been studied in this paper. Raman spectra of the first and second orders have been found to be located against the background of a luminescent halo with a maximum at ≈1250 cm$^{-1}$. Several Raman bands have been detected in the frequency range of 900–2000 cm$^{-1}$. Their frequencies were 940, 1034, 1113, 1171, 1250, 1343, 1428, 1491, 1582, 1735, 1838, and 1925 cm$^{-1}$. These bands correspond to overtone processes. We have determined that the frequencies of 1838 and 1925 cm$^{-1}$ bands are significantly higher than the exact value of the overtone frequency corresponding to the fundamental mode 4A$_1$(z)LO (864 cm$^{-1}$).

**Keywords:** lithium tantalate; double doping; Raman scattering; biphonon; laser crystal

## 1. Introduction

Nonlinear optical ferroelectric crystals of lithium niobate (LN, LiNbO$_3$) and tantalate (LT, LiTaO$_3$) are multifunctional materials widely used in modern electronics technology [1–6]. LN and LT crystals have an oxygen-octahedral structure, they are isomorphic, their unit cells contain two formula units, and the crystals are characterized by the space symmetry group R3c (C$_{3V}^6$) [2,7]. Oxygen octahedral clusters MeO$_6$ (Me–Nb$^{5+}$, Ta$^{5+}$, Li$^+$, dopant) play a crucial role in the formation of nonlinear optical and ferroelectric characteristics of LN and LT crystals [8,9]. These characteristics are of great importance for the crystals' application. Rare-earth elements (REEs)-doped LN and LT crystals are attractive as active-nonlinear laser materials due to the presence of a significant number of REE electronic transitions. The LT crystal is better as an active-nonlinear laser medium than the LN crystal, as it has a noticeably lower birefringence and higher optical damage resistance [1,3]. However, the LT crystal has a more defective crystal structure than the LN crystal, which imposes serious restrictions on the optical quality of laser materials. The optical quality of the crystals can be effectively controlled by Raman spectra [10–12].

A modern approach to the creation of diode-pumped solid-state lasers requires broadening of the absorption band of an active ion (usually a REE cation) in a laser matrix. This stabilizes the lasing process, as a stable lasing frequency of a pumping laser diode is not established instantly. The approach can be implemented in matrices based on such disordered crystals as LN and LT. At the same time, the problem of obtaining defect-free heavily doped LN and LT crystals of high optical quality with a uniform dopant distribution has not yet been finally solved.

In recent years, double-doped LN and LT crystals have attracted a lot of scientific attention. REE co-doped crystals are especially important; REE is used to generate laser radiation, and the second dopant serves to create a certain ordered active-nonlinear medium that provides optimal generation conditions, including energy transfer [13].

Growing optically and compositionally uniform LN and LT crystals with double doping is a nontrivial technological problem. Co-dopants usually have different (sometimes

strongly different) distribution coefficients $K_D$. Thus, the melt near the crystallization front can be enriched in one dopant and depleted in another during crystal growth. The composition of a doped crystal during growth can change significantly from the cone to the bottom of the crystal, which strongly decreases its compositional and optical uniformity. The characteristics of the crystal can differ markedly in its various parts. Application of natural changes in parameters during growing by the Czochralski minimizes these effects. The parameters are as follows: crystal rotation and lifting speed, temperature gradients in the melt, and growth zone. Various combinations of these parameters are also applied. In the case of the discussed crystals, there are a number of additional arrangements that can help: a special design of the thermal unit; medium crystallization rates; special preparation (small overheating, $\approx 200°$) of the melt before crystal growth; long after-growth annealing; suitable electrothermal conditions during turning of the crystals into a single-domain state.

Optical properties (absorption and luminescence spectra) of $LiTaO_3:Cr^{3+}:Nd^{3+}$ single crystals have been studied earlier in [13]. Two single-crystal samples had the same content of chromium ions and different contents of neodymium ions. The absorption spectrum was registered at room temperature. The spectrum contained two spin-allowed vibronic transitions $^4A_2$-$^4T_1$ and $^4A_2$-$^4T_2$, a spin-forbidden $^4A_2$-$^2E$ transition, and a number of $Nd^{3+}$ ions absorption bands. The luminescence spectrum had a number of bands corresponding to transitions in chromium and neodymium ions. The authors calculated the lifetime of the excited states of the chromium ion and established a nonradiative energy transfer from chromium to neodymium ions. The crystal $LiTaO_3:Cr^{3+}:Nd^{3+}$ had been determined to be a promising active nonlinear laser medium. Similar studies were carried out for other doubly doped LT crystals: $LiTaO_3:Tm^{3+}:Mg^{2+}$ [14–18], $LiTaO_3:Nd^{3+}:Mg^{2+}$ [19–27], $LiTaO_3:Mg^{2+}:Fe^{2+}$ [28–30], $LiTaO_3:In^{3+}:Nd^{3+}$ [31], $LiTaO_3:Yb^{3+}:Mg^{2+}$ [32], $LiTaO_3:Ce^{4+}:Sc^{3+}$ [33], $LiTaO_3:Zn^{2+}:Fe^{3+}$ [34], $LiTaO_3:In^{3+}:Fe^{3+}$ [35], $LiTaO_3:Ce^{4+}:Mn^{4+}$ [36], $LiTaO_3:Fe^{2+}:Ce^{4+}$ [37], and $LiTaO_3:Nd^{3+}:Yb^{3+}$ [38].

However, the articles [13–38] lack Raman spectra of the first and second orders of the studied crystals. Raman spectroscopy is a powerful tool for monitoring the state of defectiveness of LN and LT crystals, structural features of $MeO_6$ clusters, and features of the dipole ordering of structural units of the cationic sublattice. All these phenomena determine the nonlinear optical and ferroelectric properties of studied crystals.

## 2. Materials and Methods

This work presents Raman spectra of a $LiTaO_3:Cr(0.2):Nd(0.45$ wt%) single crystal. The chromium cation $Cr^{3+}$ is a photorefractive dopant that increases the sensitivity of the crystal to optical damage; the neodymium cation $Nd^{3+}$ is a nonphotorefractive dopant, reducing the effect of photorefraction. A $LiTaO_3:Cr(0.2):Nd(0.45$ wt%) crystal was grown in an argon atmosphere by the Czochralski method from a $Pt/Rh10 \varnothing 80$ mm crucible under an average axial gradient of ~12 deg/cm in the X-axis direction (X-cut) at rotation speeds of (~14 rpm) and a lifting of (~2 mm/h). The crystal growth rate was ~2.6–2.7 mm/h. Crystals were grown on a substantially modified growth facility Crystal-2 (Zavod Kristall Voroshilovgrad, USSR) equipped with an automatic crystal diameter control system. The congruent melt ([Li]/[Ta] $\approx 0.92$) was directly doped with the corresponding oxides. The studied sample was a cube with an edge length of 5 mm. The edges of the cube coincided with the direction of the main crystallophysical axes of the crystal: x, y, z. Figure 1 shows a picture of the studied sample.

The growth was completed when the weight of the $LiTaO_3:Cr(0.2):Nd(0.45$ wt%) crystal reached ~450 g. About ~25% of the total weight of the melt crystallized. The $LiTaO_3:Cr(0.2):Nd(0.45$ wt%) growth parameters of the crystal were selected based on the need to obtain a flat crystallization front, which should ensure a sufficiently high structural perfection of the crystal. The flat crystallization front was achieved by experimental selection of the following parameters: lifting speed, rod rotation speed, and temperature gradient at the crystallization front. The grown $LiTaO_3:Cr(0.2):Nd(0.45$ wt%) crystal had a flat crystallization front, $\varnothing \sim 36$ mm, and the length of the cylindrical part was $L_c \approx 40$ mm.

Dopants were introduced into the mixture as $Cr_2O_3$ and $Nd_2O_3$ oxides (concentration of impurities at a level of $<5 \times 10^{-4}$ wt%, Neva Reaktiv, Saint Petersburg, Russia). After that, the mixture was thoroughly mixed. The melt was overheated for 8 h by ~70 °C relative to the LT melting point ($T_{melt}$ = 1650 °C) before the beginning of the crystal growth. The overheating homogenizes dopants and impurities in the melt. After growing, the $LiTaO_3$:Cr(0.2):Nd(0.45 wt%) crystal was annealed at 1400 °C in a growth setup for 10 h and then cooled at a rate of ~50 deg/h. Prolonged post-growth annealing is required to homogenize the composition of the doped crystal and relieve thermal and mechanical stresses. An LT charge of congruent composition ([Li]/[Ta] $\approx$ 0.92) was synthesized using tantalum pentoxide $Ta_2O_5$ and lithium carbonate $Li_2CO_3$ (concentration of impurities at a level of $<3 \times 10^{-4}$ wt%, Solikamsk magnezium works, Solikamsk, Russia). Table 1 demonstrates the impurity composition of the LT charge and the grown $LiTaO_3$:Cr(0.2):Nd(0.45 wt%) crystal. Concentrations were determined by the spectral analysis method.

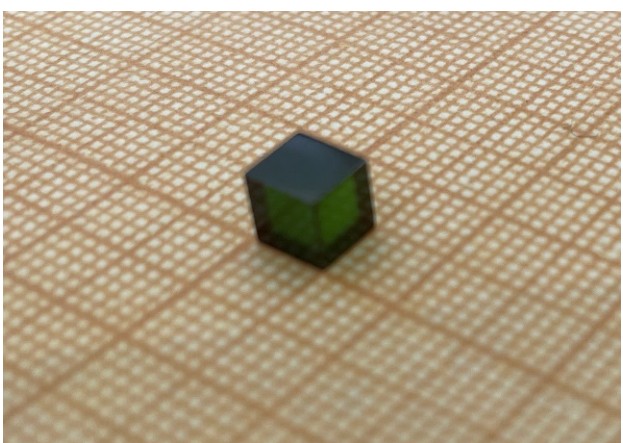

**Figure 1.** Image of the studied crystal $LiTaO_3$:Cr(0.2):Nd(0.45 wt%).

**Table 1.** Impurity composition of the initial LT charge and $LiTaO_3$:Cr(0.2):Nd(0.45 wt%) crystal.

| Impurity | Concentration, wt% | |
|---|---|---|
| | **In Charge** | **In the Crystal** |
| Mn, V, Mg, Sn | $<4 \times 10^{-4}$ | $<1 \times 10^{-4}$ |
| Pb, Ni | $<5 \times 10^{-4}$ | $<4 \times 10^{-4}$ |
| Co, Mo | $<3 \times 10^{-4}$ | $<4 \times 10^{-4}$ |
| Si, Fe | $<3 \times 10^{-4}$ | $<5 \times 10^{-4}$ |
| Ti | $<3 \times 10^{-4}$ | $<4 \times 10^{-4}$ |
| Al | $<8 \times 10^{-4}$ | $<7 \times 10^{-4}$ |
| Zr | $<5 \times 10^{-4}$ | $<1 \times 10^{-3}$ |
| Ca | $<4 \times 10^{-4}$ | $<6 \times 10^{-4}$ |
| Te, Sb | $<5 \times 10^{-4}$ | $<3 \times 10^{-4}$ |
| Bi | $<4 \times 10^{-4}$ | $<3 \times 10^{-4}$ |
| Rh | $<1 \times 10^{-4}$ | $<1 \times 10^{-2}$ |

The Cr and Nd concentrations in $LiTaO_3$:Cr(0.2):Nd(0.45 wt%) crystal were determined by atomic emission spectrometry on the spectrometer ICPE-9000 (Shimadzu, Kyoto, Japan, 2011). The data were controlled by X-ray fluorescence analysis on a Spectroscan MAKS-GV (Spectron, Saint Petersburg, Russia).

Raman spectra were excited by a cw semiconductor laser from a BWS465-785H i-Raman Plus spectrometer (B&W Tek, Plainsboro city, NJ, USA) with a wavelength of $\lambda$ = 785 nm and a power of 200 mW. Exciting infrared radiation ensured the absence of the effect of photorefraction (optical damage). Photorefraction is a photoinduced change in the refractive indices of the crystal. Exciting laser radiation was introduced into the first channel of a two-channel optical fiber, excited the optical fiber, and was focused by

two lenses onto the sample surface. The laser beam fell along or perpendicular to the polar Z-axis of the studied crystal. The focal waist was located at the center of the crystal. The scattered light was collected by the same lenses in the opposite direction and was introduced into the second channel of the light guide. After that, radiation hit a selective light filter that cuts off the exciting radiation. Finally, the Raman signal fell on the slit of a minispectrometer. The BWS465-785H model has a multi-element receiver; it can record the Raman spectrum in the range of 50–2850 cm$^{-1}$ with a spectral resolution of $\approx$3.5 cm$^{-1}$.

### 3. Results and Discussion

Figure 2 shows the recorded Raman spectra in three mutually perpendicular directions of the studied sample: $x(zz, zy)\overline{x}$, $y(zz, zx)\overline{y}$, and $z(xx, yy, xy)\overline{z}$. The first-order Raman spectrum of a nominally pure LT crystal is located in the range of 140–1000 cm$^{-1}$, with these bands corresponding to fundamental vibrations of the crystal lattice [39–41]. Bands are absent from the region above 1000 cm$^{-1}$ of a nominally pure LT crystal. Figure 2 demonstrates that the spectrum of the LiTaO$_3$:Cr(0.2):Nd(0.45 wt %) crystal in the region above 1000 cm$^{-1}$ exhibits weak second-order spectrum bands corresponding to overtone processes.

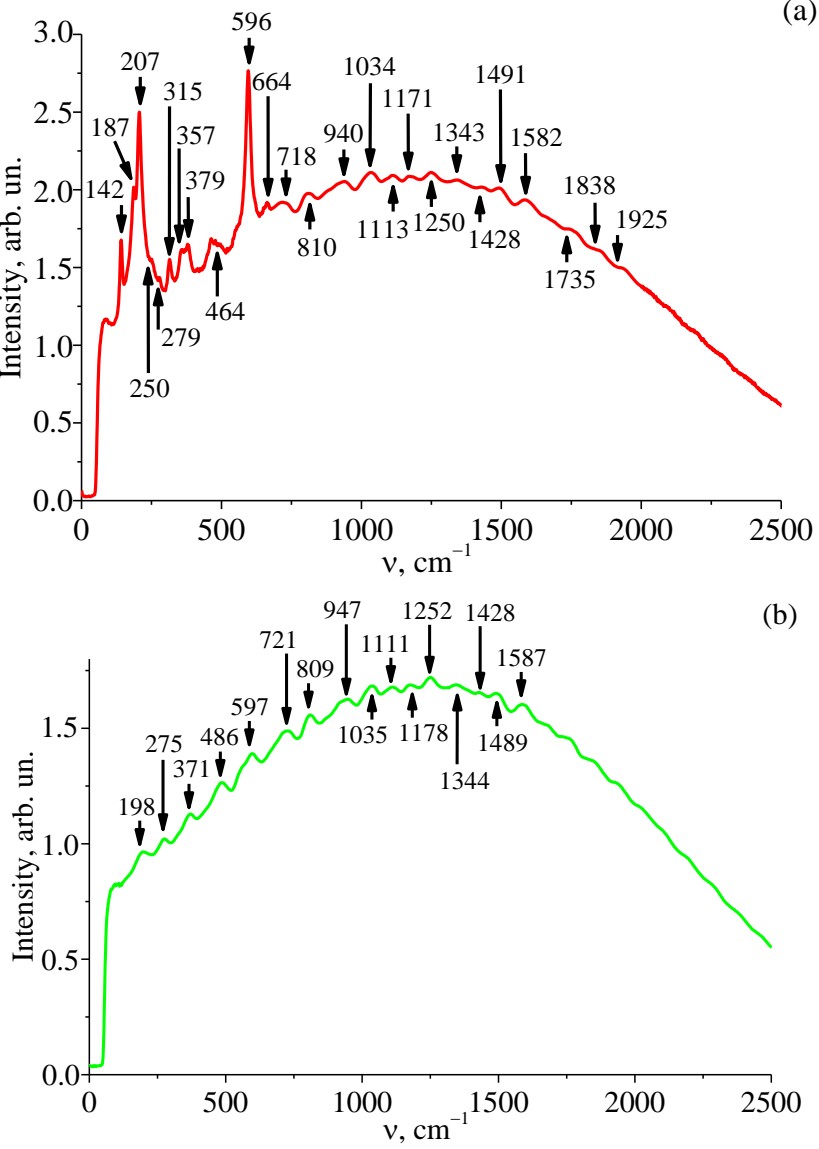

**Figure 2.** *Cont.*

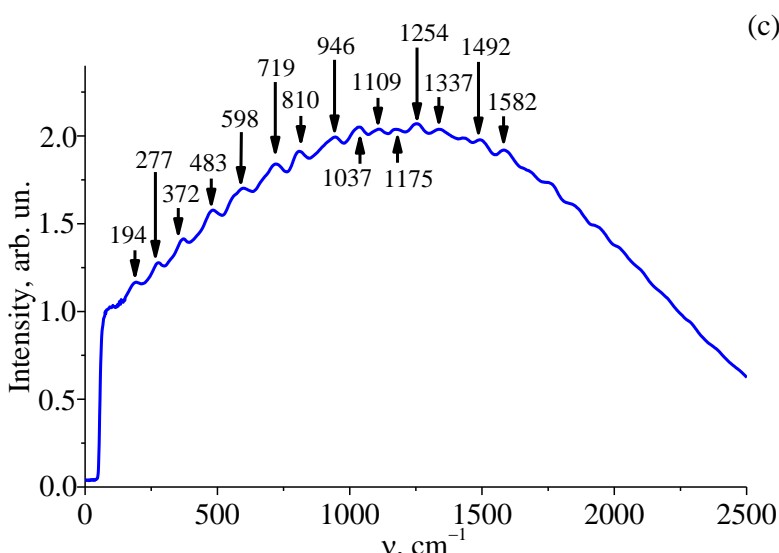

**Figure 2.** Raman spectra in the region of fundamental modes for 180-degree scattering (backscattering) of LiTaO₃:Cr(0.2):Nd(0.45 wt%) crystal with scattering geometries $x(zz, zy)\bar{x}$ (**a**), $y(zz, zx)\bar{y}$ (**b**), and $z(xx, yy, xy)\bar{z}$ (**c**).

Figure 2 also shows that the Raman spectra of the first and second orders of the LiTaO₃:Cr(0.2):Nd(0.45 wt %) crystal are located against the background of a luminescent halo with a maximum at $\approx 1250$ cm$^{-1}$. LiTaO₃:Cr and LiTaO₃:Cr,Nd crystals' luminescent spectra have a maximum near 890 nm at 5 K [13]. The position of the luminescence maximum is close to the position of the luminescent halo observed in our Raman spectra. The Raman spectrum in the scattering geometry $x(zz, zy)\bar{x}$ contains intense bands corresponding to the longitudinal fundamental polar modes of 1A₁(z)-, 3A₁(z)-, and 4A₁(z)-types with polarization along the Z-axis. In addition, the manifestation of doubly degenerate E(x,y) oscillations is possible in the same scattering geometry in accordance with the form of the Raman tensor [42]. It should be noted that for this geometry, there is no fully symmetric mode 4A₁(z)LO in the Raman spectrum. This fact is due to the following reasons. This mode has a low intensity in a pure (undoped) LT crystal. Probably, the wide luminescent halo and the second-order Raman lines overlap in our case. Against this background, the fully symmetric 4A₁(z)LO mode is not visible in the recorded Raman spectrum. In the 800–2000 cm$^{-1}$ range, we determined bands with the frequencies 810, 940, 1034, 1113, 1171, 1250, 1343, 1428, 1491, 1582, 1735, 1838, and 1925 cm$^{-1}$. The bands do not have a strict structure and their spectral width is several tens of cm$^{-1}$. Thus, they cannot be attributed to neodymium ions luminescence. This is why we believe that the bands belong to the second-order spectrum. Note that the frequencies of 1838 and 1925 cm$^{-1}$ bands are much greater than the overtone value; the overtone belongs to a full-symmetry fundamental mode 4A₁(z)LO (864 cm$^{-1}$).

Raman spectra of the other two geometries ($y(zz, zx)\bar{y}$ and $z(xx, yy, xy)\bar{z}$) have a completely different form (see Figure 2b,c). Weak first- and second-order Raman bands are observed against the background of a wide luminescent halo. Moreover, these spectra differ from the corresponding Raman spectra of nominally pure LT. The reason for the observed difference is likely caused by changes in the defective structure due to a double-doping of the crystal by chromium and neodymium. This is caused by the location of dopants in the sites of intrinsic structure cations; thus, the general alteration of intrinsic cations along the polar axis changes. In this case, the interaction between the defect structure of the crystal and the exciting laser radiation should change.

Table 2 lists the values of all the LT frequencies of the main fundamental polar modes measured in this work; the bands attribution was made due to data available from [43] (Attribution 1) and [44] (Attribution 2). An alternative attribution of some Raman bands is

shown in parentheses. Table 2 reveals that frequencies of transverse and longitudinal modes are very different, which is typical for fundamental polar vibrations in noncentrosymmetric crystals. These differences cannot be due to the effect of photorefraction, as the spectra were excited by IR radiation with a wavelength of 785 nm.

**Table 2.** Frequencies of the transverse (TO) and longitudinal (LO) polar modes of the LiTaO$_3$:Cr(0.2):Nd(0.45 wt%) crystal obtained in this work and their attribution.

| ν, cm$^{-1}$ | Attribution 1 | Attribution 2 |
|:---:|:---:|:---:|
| 142 | 1E(x,y)TO | 1E(x,y) |
| 187 | 1E(x,y)LO | 2E(x,y) |
| 207 | 2E(x,y)LO (1A$_1$(z)TO) | 3E(x,y)TO |
| 250 | 3E(x,y)TO | 4E(x,y)TO |
| 279 | 3E(x,y)LO | |
| 315 | 4E(x,y)TO | 5E(x,y)TO |
| 357 | 3A$_1$(z)TO | |
| 379 | 5E(x,y)LO | 6E(x,y)TO |
| 464 | 7E(x,y)TO | 7E(x,y)TO |
| 596 | 4A$_1$(z)TO (8E(x,y)TO) | 8E(x,y)TO |
| 664 | 8E(x,y)LO (9E(x,y)TO) | 9E(x,y)TO |

The scheme of the formation of bound states of phonons in crystals was considered in theoretical works [45–47]. In accordance with these works, the density of two-phonon states $\rho_2(\omega)$ was calculated using one-phonon Green's functions $D_1(k,\omega)$ according to Equations (1)–(5):

$$D_1\left(\vec{k},\omega\right) = \frac{\omega\left(\vec{k}\right)}{2}\left[\frac{1}{\omega - \omega\left(\vec{k}\right) + \frac{1}{2}i\Gamma} - \frac{1}{\omega + \omega\left(\vec{k}\right) - \frac{1}{2}i\Gamma}\right] \qquad (1)$$

$$F(\omega) = \frac{i}{(2\pi)^4}\int d^3\vec{k}\int D_1\left(\vec{k},\omega - \omega'\right)D_1\left(\vec{k},\omega\right)d\omega' \qquad (2)$$

In the quasi-Newtonian approximation of the dispersion law for optical phonons with a frequency $\omega_0$ near the center of the Brillouin zone, the density of single-particle states of phonons $\rho_1(\omega)$ satisfies the following relation:

$$\rho_1(\omega) = a\sqrt{\omega_0 - \omega} \qquad (3)$$

where $a = -\frac{\omega_0\sqrt{2\omega_0}}{2\pi^2 s^3}$ and the constant s is the speed of sound in an LT crystal. As a result, for the function F($\omega$), we obtain:

$$F(\omega) = \frac{1}{4}\omega_0^2 a\int_0^\Delta \frac{\sqrt{\omega'}}{\omega - 2(\omega_0 - \omega') + i\Gamma}d\omega' \qquad (4)$$

where $\Delta$ is the phonon frequency range taken into account in the integration. It is important to note that the function F($\omega$) depends on the effective mass of the quasiparticle and on the damping constant $\Gamma$ associated with the inverse lifetime of the quasiparticle $\tau = 1/\Gamma$. The second-order Raman intensity is proportional to the density of two-phonon states $\rho_2(\omega)$:

$$\rho_2(k,\omega) \approx -\frac{2}{\pi\omega_0^2}\frac{\mathrm{Im}F(\omega)}{\left[1 - \frac{1}{2}g_4\mathrm{Re}F(\omega)\right]^2 + \left[\frac{1}{2}g_4\mathrm{Im}F(\omega)\right]^2} \qquad (5)$$

The literature [43,44,48] states that the fully symmetric $4A_1(z)LO$ mode of nominally pure LT has a strong dispersion dependence and the highest frequency of 864 cm$^{-1}$. Figure 3 shows the results of comparing the calculated spectral distribution of the Raman in the overtone region $2\nu_0 = 1728$ cm$^{-1}$ and the experimentally observed intensity distribution. The dotted arrow in this figure indicates the exact position of the $2\nu_0$ overtone.

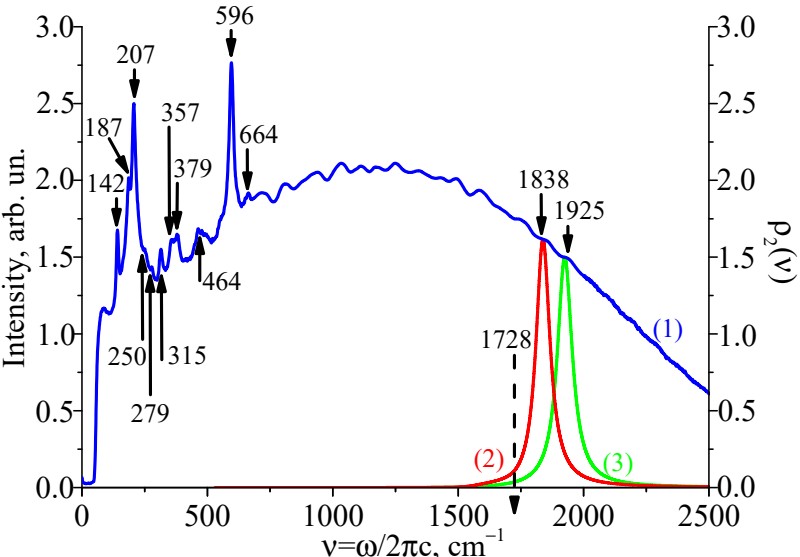

**Figure 3.** Comparison of the spectral Raman intensity of the LiTaO$_3$:Cr(0.2):Nd(0.45 wt%) crystal (curve 1) in the overtone region with the calculated dependences of the density of two-phonon states $\rho_2(\nu)$ (curves 2 and 3).

Figure 3 confirms that the 1838 cm$^{-1}$ band (curve 2) is satisfactorily approximated by the bound state of the $4A_1(z)$ polar mode with a frequency of 864 cm$^{-1}$ with the following parameter values: s = 1000, $\Delta = 0.1\omega_0$, $g_4 = 24.35 \cdot 10^{-46}$, and $\Gamma = 7.54 \cdot 10^{12}$. For the 1925 cm$^{-1}$ band (curve 3), satisfactory agreement is achieved at s = 1000, $\Delta = 0.1\omega_0$, $g_4 = 16.43 \cdot 10^{-46}$, and $\Gamma = 7.54 \cdot 10^{12}$. The value of the constant s is close in order of magnitude to the speed of sound in LT. The value of $\Delta$ was chosen based on the form of the dispersion curve of optical phonons $4A_1(z)$. The damping constant $\Gamma$ corresponds to the experimental width of the Raman band of these phonons.

## 4. Conclusions

Full Raman spectra of a LiTaO$_3$:Cr(0.2):Nd(0.45 wt%) crystal were recorded in the backscattering geometry; their interpretation was given. The spectrum contains a number of fundamental Raman bands and overtone weak bands in the range 800–2000 cm$^{-1}$. Two bands with frequencies of 1838 and 1925 cm$^{-1}$, belonging to the second-order spectrum, exceed the exact value of the overtone of the $4A_1(z)$ mode.

In order to establish the optimal composition and optimal physical characteristics of double-doped LiTaO$_3$:Cr:Nd crystals suitable for active nonlinear laser material, we will perform complex studies of the crystals using photoluminescence, photoinduced, and Raman scattering, including stimulated Raman scattering, upon excitation with various laser radiation in the UV, visible, and IR ranges. It is also necessary to establish and study the conditions for observing laser generation in these crystals.

**Author Contributions:** Conceptualization, N.S.; validation, N.S.; investigation, A.P.; data curation, A.P.; writing—original draft preparation, N.S. and A.P.; writing—review and editing, M.P.; visualization, M.P.; supervision, M.P. All authors have read and agreed to the published version of the manuscript.

**Funding:** This work was supported by the Ministry of Higher Education Russian Federation scientific topic 0186-2022-0002 (registration FMEZ-2022-0016) and the Russian Foundation for Basic Research grant 20-52-04001 Bel_mol_a.

**Institutional Review Board Statement:** Not applicable.

**Informed Consent Statement:** Not applicable.

**Data Availability Statement:** Data on this research will be available from the corresponding author, N.S., on a reasonable request.

**Conflicts of Interest:** The authors declare no conflict of interest.

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
