# Peer review of "Raman Scattering in a Double-Doped Single Crystal LiTaO3:Cr(0.2):Nd(0.45 wt%)"

_photonics, doi:10.3390/photonics9100712_

Round 1

Reviewer 1 Report

The manuscript is about first order and second order Raman scattering in a LiTaO3:Cr:Nd single crystal.

The interpretation of second order spectra must be reformulated and made accessible to a wide audience.

Generally speaking, the manuscript is not accurately written. I recommend a thorough revision before it is considered for publication.

In the following, I list some specific comments.

- Line 11: "for the first time" sort of statements should be avoided.

- Line 28: The acronym "REE" must be defined.

- Lines 79 - 81:

"It is obvious that in the more perfect LN and LT crystals structure, the second-order Raman spectrum should be less intense [40]: the second-order spectrum should be absent if the structure is ideal."

None of the two statements in lines 79 - 81 is "obvious". The authors must rewrite these lines, and justify all the statements. Ref. 40 is a paper written in Russian. Most readers, including myself, do not understand Russian. The authors must support all their statements by adding appropriate references to articles written in English.

Line 143: According to manufacturer's specification spectral resolution is about 4.5 cm^-1, which is not as good as stated by the authors.

Throughout the manuscript the authors typed numbers, symbols, and units of measurements neglecting to fix superscripts and subscripts. This has particularly deplorable effects in the equation in line 146, that needs to be decrypted. 

Line 146 - "the wave vector of polar excitations appearing in the first-order Raman spectra was comparable with the wave vector of the exciting radiation: kp≈2klas≈105 cm-1." The use of the same unit (cm-1) for wave vectors and for Delta(frequency), or Raman shift may be confusing. The authors should add some clarification.

Lines 173 - 174: The identification of overtone processes with bound states of optical phonons must be clarified and justified.

Line 201: The symbol s must be defined.

Lines 229 - 230: See comment to line 11.

Axes labels in figure 2 should be more clear (figure 3 is better).

Author Response

Reviewer 1 wrote:

The manuscript is about first order and second order Raman scattering in a LiTaO3:Cr:Nd single crystal.

The interpretation of second order spectra must be reformulated and made accessible to a wide audience.

Generally speaking, the manuscript is not accurately written. I recommend a thorough revision before it is considered for publication.

Answer:

Dear reviewer!

We are grateful for the time and effort you have kindly dedicated to our paper. Changes made due to your valuable comments were little but crucial. Please note that changes in the manuscript made due to your comments are highlighted in yellow, due to the other reviewer – blue. Changes in English are written in purple letters.

Reviewer 1 wrote:

In the following, I list some specific comments.

- Line 11: "for the first time" sort of statements should be avoided.

Answer:

We have erased this statement from the text.

Reviewer 1 wrote:

- Line 28: The acronym "REE" must be defined.

Answer:

The acronym is deciphered at its first mention.

Reviewer 1 wrote:

- Lines 79 - 81:

"It is obvious that in the more perfect LN and LT crystals structure, the second-order Raman spectrum should be less intense [40]: the second-order spectrum should be absent if the structure is ideal."

None of the two statements in lines 79 - 81 is "obvious". The authors must rewrite these lines, and justify all the statements. Ref. 40 is a paper written in Russian. Most readers, including myself, do not understand Russian. The authors must support all their statements by adding appropriate references to articles written in English.

Answer:

We have deleted the doubtful piece and the reference. Now there are 49 references, not 50 as was before.

Reviewer 1 wrote:

Line 143: According to manufacturer's specification spectral resolution is about 4.5 cm^-1, which is not as good as stated by the authors.

Answer:

We have a model with the range 50-2850 cm-1; a resolution of 3.5 cm-1 is available in it. The manufacturer specifies exactly this error for a newer spectrometer model.

Reviewer 1 wrote:

Throughout the manuscript the authors typed numbers, symbols, and units of measurements neglecting to fix superscripts and subscripts. This has particularly deplorable effects in the equation in line 146, that needs to be decrypted. 

Line 146 - "the wave vector of polar excitations appearing in the first-order Raman spectra was comparable with the wave vector of the exciting radiation: kp≈2klas≈105 cm-1." The use of the same unit (cm-1) for wave vectors and for Delta(frequency), or Raman shift may be confusing. The authors should add some clarification.

Answer:

In the new version of the manuscript, the upper and lower indices have been adjusted. The text starting on line 146 has been removed from the new version of the manuscript.

Reviewer 1 wrote:

Lines 173 - 174: The identification of overtone processes with bound states of optical phonons must be clarified and justified.

Answer:

A short discussion of overtone processes has been added to the new version of the article.

Reviewer 1 wrote:

Line 201: The symbol s must be defined.

Answer:

Explanation is added for parameter s.

Reviewer 1 wrote:

Lines 229 - 230: See comment to line 11.

Answer:

We have erased this statement from the text.

Reviewer 1 wrote:

Axes labels in figure 2 should be more clear (figure 3 is better).

Answer:

Figures 2 a, b, с have been replaced; axis labels are clearer and font is larger.

Reviewer 2 Report

Nonlinear optical ferroelectric crystals of lithium niobate (LN, LiNbO3) and tantalate (LT, LiTaO3) are multifunctional materials widely used in modern electronic technology. In recent years, the double doping of LN and LT crystals has attracted much attention of scientists. Crystals doped with REE are especially important; REE are used to generate laser radiation, the second dopant serves to create an effective absorption of exciting radiation with further transfer of excitation to REE. LN and LT crystals doped with REE are attractive as a material for active-nonlinear lasers due to the presence of a significant number of electronic REE transitions.

There are inaccuracies in the work that need to be corrected:

1. The edges of the cube coincided with the direction of the main crystallographic axes of the crystal x, y, z. Fig. 1 shows a picture of the studied sample - line 93-94,

however, the symmetry of the crystal is trigonal, the two crystallographic axes are at an angle of 120 degrees, not 90. We are talking about crystallophysical axes. At the same time, the photo shows a sample cut out in the form of a cube.

2. In the initial reagents, the concentration of impurity is at level of < 3· 10-4 wt%, and in the crystal it increases for many impurities. What is the reason? (especially for Rh, where the concentration in the crystal increases by two orders of magnitude compared to the charge).

3. The authors give the crystal composition as LiTaO3:Cr(0.2):Nd(0.45 wt%), but do not give methods for determining the weight concentrations of dopants.

4. Line 73 does not specify the valence of iron LiTaO3:Fe:Ce4+ [38]

5. There is no need to give two figures with crystals. One thing is enough: better (b).

6. Section 2 (Materials and methods) does not provide the melting point of the crystal, only deviations from it. It is necessary to specify.

7. There is a lot of talk about using Raman spectroscopy to characterize a crystal, but nothing is really given!

8. Grammatical errors (typos):

Line27 grEat

Line136 exCited

Author Response

Reviewer 2 wrote:

Nonlinear optical ferroelectric crystals of lithium niobate (LN, LiNbO3) and tantalate (LT, LiTaO3) are multifunctional materials widely used in modern electronic technology. In recent years, the double doping of LN and LT crystals has attracted much attention of scientists. Crystals doped with REE are especially important; REE are used to generate laser radiation, the second dopant serves to create an effective absorption of exciting radiation with further transfer of excitation to REE. LN and LT crystals doped with REE are attractive as a material for active-nonlinear lasers due to the presence of a significant number of electronic REE transitions.

Answer:

Dear reviewer!

We appreciate your attention during reading of our manuscript. Answer to your valuable recommendations helped to improve our paper a lot. Please note that changes in the manuscript made due to your comments are highlighted in blue, due to the other reviewer – yellow. Changes in English are written in purple letters.

Reviewer 2 wrote:

There are inaccuracies in the work that need to be corrected:

  1. The edges of the cube coincided with the direction of the main crystallographic axes of the crystal x, y, z. Fig. 1 shows a picture of the studied sample - line 93-94, however, the symmetry of the crystal is trigonal, the two crystallographic axes are at an angle of 120 degrees, not 90. We are talking about crystallophysical axes. At the same time, the photo shows a sample cut out in the form of a cube.

Answer:

You are absolutely right - these are crystallophysical axes. An appropriate correction has been made to the text of the article.

Reviewer 2 wrote:

  1. In the initial reagents, the concentration of impurity is at level of < 3· 10-4 wt%, and in the crystal it increases for many impurities. What is the reason? (especially for Rh, where the concentration in the crystal increases by two orders of magnitude compared to the charge).

Answer:

Some impurities were filtered out, some impurities undergo a regular increase in the concentration in the crystal (Table 1). For example, a slight increase in the concentration of zirconium and calcium is due to the fact that zirconium ceramics (containing calcium) was used in the design of the thermal unit during crystal growth. A noticeable increase in the Rh concentration in the crystal is due to the fact that platinum crucibles containing rhodium (Pt/Rh10%) were used for its growth. Rhodium is added to platinum to increase the melting point of the melt.

Reviewer 2 wrote:

  1. The authors give the crystal composition as LiTaO3:Cr(0.2):Nd(0.45 wt%), but do not give methods for determining the weight concentrations of dopants.

Answer:

This is our omission. In the revised text of the article, this information is given in the Experiment section:

“Cr and Nd concentration in LiTaO3:Cr(0.2):Nd(0.45 wt%) crystal was determined by by atomic emission spectrometry on spectrometer ICPE-9000 (Shimadzu, Japan, Kyoto, 2011). The data were controlled by X-ray fluorescence analysis on Spectroscan MAKS-GV (Spectron, Russia, Saint Petersburg).”

Reviewer 2 wrote:

  1. Line 73 does not specify the valence of iron LiTaO3:Fe:Ce4+ [38]

Answer:

In the new version of the manuscript, on line 73, the valency of iron is added.

Reviewer 2 wrote:

  1. There is no need to give two figures with crystals. One thing is enough: better (b).

Answer:

In the new version of the article, one photo of the test sample was left; As you recommended, it is b.

Reviewer 2 wrote:

  1. Section 2 (Materials and methods) does not provide the melting point of the crystal, only deviations from it. It is necessary to specify.

Answer:

Thank you for this recommendation. This is important, but we did not add the actual melting temperature. The text has been changed and now this sentence looks like this: “The melt was overheated for 8 h by ~70 °С relative to the LT melting point (Тmelt = 1650°С) before the beginning of the crystal growth.”

Reviewer 2 wrote:

  1. There is a lot of talk about using Raman spectroscopy to characterize a crystal, but nothing is really given!

Answer:

In this work, the first and second order Raman spectra of this exact crystal are recorded and interpreted for the first time. As indicated in the conclusions, further work will be related to the optimization of the concentration of dopants. New Raman spectra will be obtained for such crystals.

Reviewer 2 wrote:

  1. Grammatical errors (typos):

Line27 grEat

Line136 exCited

Answer:

The typos are corrected.

Round 2

Reviewer 1 Report

(a)  -  The revised manuscript is still unclear about two points:

1 - Why the 4A1 (864 cm-1) mode is absent in the reported Raman spectra

2 - Reference(s) to 4A1 (864 cm-1) mode frequency and attribution are missing

(b)  -   As I tried to recommend also in my first report, axes labels in Figures 2 and 3 must be uniformed. I strongly suggest to adopt the vertical label of Figure 2 and the horizontal label of Figure 3.

Author Response

Reviewer 1 wrote:

(a) The revised manuscript is still unclear about two points:

1 - Why the 4A1 (864 cm-1) mode is absent in the reported Raman spectra

2 - Reference(s) to 4A1 (864 cm-1) mode frequency and attribution are missing

(b)  -   As I tried to recommend also in my first report, axes labels in Figures 2 and 3 must be uniformed. I strongly suggest to adopt the vertical label of Figure 2 and the horizontal label of Figure 3.

Answer:

Dear reviewer!

Thank you for your comments. Your attention to details in our Manuscript helped us improve our work.

(a)

  1. The literature [1–4] says: for the x(zz, zy) ̅x scattering geometry, the 4A1(z)LO fully symmetric mode has a low intensity in a pure (undoped) lithium tantalate crystal. In our work, we were also expecting a low intensity of this mode in this geometry. Probably, the wide luminescent halo and the second-order Raman lines overlap. Against this background, the fully symmetric 4A1(z)LO mode is not visible in the recorded Raman spectrum.

[1] A. F. Penna, A. Chaves, P. da R. Andrade, S. P. S. Porto, Phys. Rev. 13, 4907 (1976).

[2] C. Rapits, Phys. Rev. B 38, 10007 (1988).

[3] S. Sanna, S. Neufeld, M. Rüsing, G. Berth, A. Zrenner, W. G. Schmidt, Phys. Rev. B 91, 224302 (2015).

[4] V. S. Gorelik, S. D. Abdurakhmonov, N. V. Sidorov, M. N. Palatnikov, Inorg. Mater. 55, 524 (2019).

The reason for y(zz, zx) ̅y and z(xx, yy,xy) ̅z geometries the reason is different. Defective structure of doubly-doped crystal is strongly changed. This is caused by location of dopants in the sites of intrinsic structure cations, thus, the general alteration of intrinsic cations along the polar axis changes. In this case, the interaction between the defect structure of the crystal and the exciting laser radiation should change. Unfortunately, we don't know the exact reason.

  1. The article contains references to works [39-41, 43, 44, 48], which describe the vibrational mode 4A1(z) and its frequencies

(b) In the new version of the article in Fig. 2 and 3, the same step is made on both axes.

Reviewer 2 Report

I think the work may be accepted.

Author Response

Reviewer 2 wrote:

I think the work may be accepted.

Answer:

Dear reviewer!

Thank you very much! Your priceless comments have helped us to improve our manuscript. It was a pleasure to work with you.